# Distributed Network of Adaptive and Self-Reconfigurable Active Vision Systems

**Shashank \* and Indu Sreedevi**

Department of Electronics and Communication Engineering, Delhi Technological University, Delhi 110042, India
\* Correspondence: shashank.ece.dtu@gmail.com or shashank_2k17phdec05@dtu.ac.in

**Abstract:** The performance of a computer vision system depends on the accuracy of visual information extracted by the sensors and the system's visual-processing capabilities. To derive optimum information from the sensed data, the system must be capable of identifying objects of interest (OOIs) and activities in the scene. Active vision systems intend to capture OOIs with the highest possible resolution to extract the optimum visual information by calibrating the configuration spaces of the cameras. As the data processing and reconfiguration of cameras are interdependent, it becomes very challenging for advanced active vision systems to perform in real time. Due to limited computational resources, model-based asymmetric active vision systems only work in known conditions and fail miserably in unforeseen conditions. Symmetric/asymmetric systems employing artificial intelligence, while they manage to tackle unforeseen environments, require iterative training and thus are not reliable for real-time applications. Thus, the contemporary symmetric/asymmetric reconfiguration systems proposed to obtain optimum configuration spaces of sensors for accurate activity tracking and scene understanding may not be adequate to tackle unforeseen conditions in real time. To address this problem, this article presents an adaptive self-reconfiguration (ASR) framework for active vision systems operating co-operatively in a distributed blockchain network. The ASR framework enables active vision systems to share their derived learning about an activity or an unforeseen environment, which learning can be utilized by other active vision systems in the network, thus lowering the time needed for learning and adaptation to new conditions. Further, as the learning duration is reduced, the duration of the reconfiguration of the cameras is also reduced, yielding better performance in terms of understanding of a scene. The ASR framework enables resource and data sharing in a distributed network of active vision systems and outperforms state-of-the-art active vision systems in terms of accuracy and latency, making it ideal for real-time applications.

**Keywords:** active vision; self-adaptation; self-reconfiguration; smart camera network





## 1. Introduction

A smart camera network (SCN), defined by Reisslein et al. [1], is a real-time distributed embedded system that is configured to perform computer-vision tasks by processing sensor data obtained from a plurality of cameras via cooperative sensing. The smart camera network is generally deployed to perform complex computer-vision tasks that require more than one camera to extract visual information. Visual surveillance of large areas [2], complex sports analytics [3], situation cognizance [4], tele-immersion, automated driverless vehicles [5], ambient assisted living [6], computer-vision-based disaster management, robotic vision, and other applications rely on SCNs to achieve desired functionalities. The smart camera network aims to co-operatively fuse sensor data to develop scene understanding and further associates scene understanding with a control system configured to achieve the overall functionality of a computer vision system.

The advancement in computer-vision applications has taken a tremendous leap in the last decade. According to a survey reported in [2], the growth of computer-vision applications is estimated at a compound annual growth rate (CAGR) of 7.6% from 2020

to 2027. Computer vision systems [7] aim to leverage human efforts by developing an understanding of events through the processing of data obtained from a number of sensors (hereinafter interchangeably referred to as "cameras" or "camera sensors"). Further, based on understanding, most of the smart computer vision systems aim to enable the operation of control systems to deliver a desired functionality automatically without any human interaction. An ideal computer vison system should extract data from the activities detected by the sensors, generate an understanding based on processing of the data, and provide optimal functionality through an appropriate control system.

Computer vision systems [7] rely on the processing of digital images or videos obtained from one or more cameras to obtain an understanding of an activity or an event. Understanding of the activity or event highly depends on the quality of data in terms of information about objects of interest. To obtain optimum information about an OOI from captured data, the cameras must be configured to capture OOIs with the highest possible resolution. In some respects, it can be concluded that the objects of interest must be captured as close to the center of the camera's field of view as possible.

A computer vision system [7] needs to process the data to identify an OOI and critical activities in the scene to derive scene understanding, based on which the cameras can be reconfigured to capture the OOI in the center of the FOV. Thus, data processing and camera calibration are interdependent. A system capable of reconfiguring the parameters of the cameras to manipulate the viewpoints of the cameras in order to investigate the environment and obtain better information from it is known as an active vision system. As reconfiguration of the cameras and the processing performance of the active vision system are interdependent, designing advanced active vision systems employing a network of cameras co-operatively working towards a desired functionality in real time is very challenging.

Traditional model-based active vision systems have limited computational resources, due to which they can only identify activities in known environments for which they are designed and fail badly to identify and understand new activities in unforeseen conditions. Active vision systems employing artificial intelligence (AI) manage to tackle unforeseen conditions; however, due to iterative training processes, they are not reliable for real-time applications. The impacts of unforeseen conditions and uncertainties in computer vision systems are presented in [2].

To address the abovementioned problem, this article showcases an adaptive self-reconfiguration (ASR) framework for active vision systems operating co-operatively in a distributed blockchain network. The ASR framework facilitates active vision systems to share information about their learning towards new activities and unforeseen environments with other systems in the distributed network. The information shared can be utilized by any system in the distributed blockchain network to tackle an identical condition and thus saves a lot of time which would otherwise have been taken up with iterative model training. Further, as the learning duration is reduced, the duration of the reconfiguration of the cameras is also reduced, thus yielding better performance in terms of understanding a scene.

To develop a good understanding of the abovementioned reconfiguration problem and its impact on the performance of an active vision system, this article primarily provides a detailed discussion of the challenges in developing an active vision system at different operational levels. We further provide an extended survey of various systems and methods proposed for addressing the challenges. Additionally, this article highlights a trend in the state-of-the-art systems and methods proposed to address the active vision challenges and showcases a common threat for most of the contemporary solutions. This article further provides definitions of the concepts of self-reconfiguration and self-adaptation and presents a role for self-reconfiguration and self-adaptation in the enhancement of the performance of active vision systems. Finally, this article provides an adaptive self-reconfiguration (ASR) framework to enhance the performance of active vision systems and their applications.

A computer vision system generally includes one or more camera sensors, software or circuitry for data processing, and a control system. The camera sensors are configured for co-operative sensing in the operating environment and gathering raw sensor data. The raw

sensor data are fused to obtain pre-processed visual information that is utilized for visual processing. The visual processing includes several steps to process and furnish the sensor data and obtain a visual understanding. A control signal is generated based on the visual understanding and is transmitted to the control system. Based on the control signal received, the control system actuates one or more components of the system such that the desired functionality of the system is achieved. The abovementioned process is depicted in Figure 1.

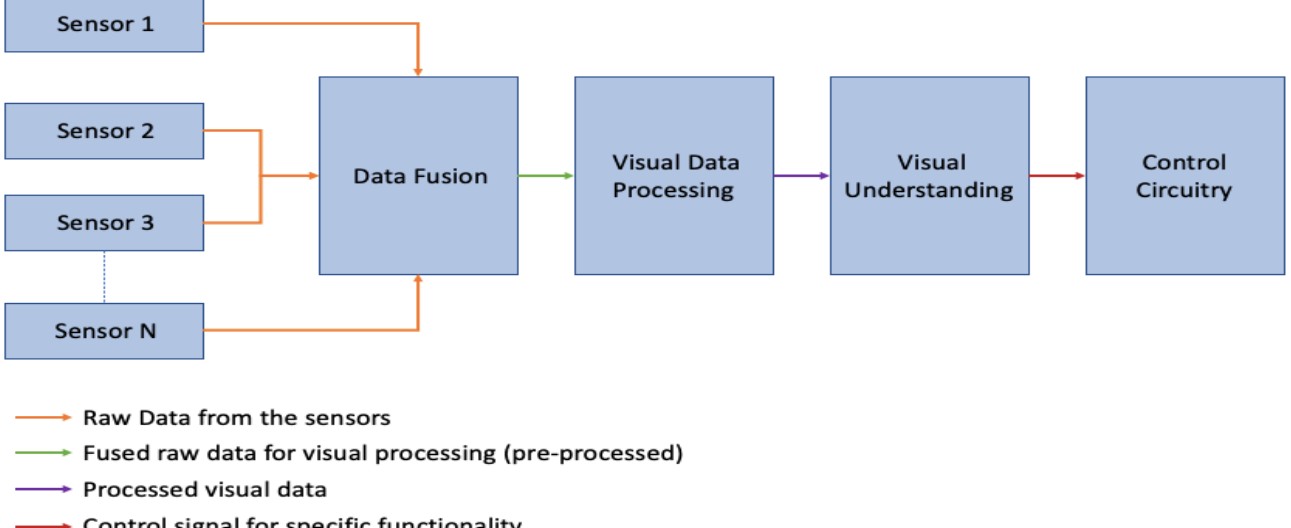

**Figure 1.** A general process flow of a computer vision system employing a smart camera network.

A multi-camera active vision system [8] (hereinafter interchangeably referred to as an "active vision system") is basically a computer vision system that employs an SCN with the capability of altering the viewpoints of the sensor nodes, thereby yielding better data for processing. The operation of active vision systems can be differentiated into two levels: a deployment level (i.e., for data extraction) and a processing level (i.e., where the visual processing takes place to obtain an understanding). The challenges associated with the deployment of an active vision system employing one or more SCNs can also be classified into two categories. Challenges of the first type [9] are associated with reconfiguration of the sensor nodes to obtain sensor data bearing optimal information, relative to each camera's resource limitations. Challenges of the second type [2] are associated with the data-processing level of operation, occurring at the time of processing the data to develop an understanding of a scene while keeping the computational complexity as low as possible. Thus, an efficient active vision system demands efforts at both hardware and software levels of deployment. Detailed reviews of the challenges at the data-processing level and the sensor-calibration level of active vision systems are presented in [2] and [9], respectively. A multi-tier taxonomy of challenges for active vision systems employing camera networks on the basis of operational classification is shown in Figure 2.

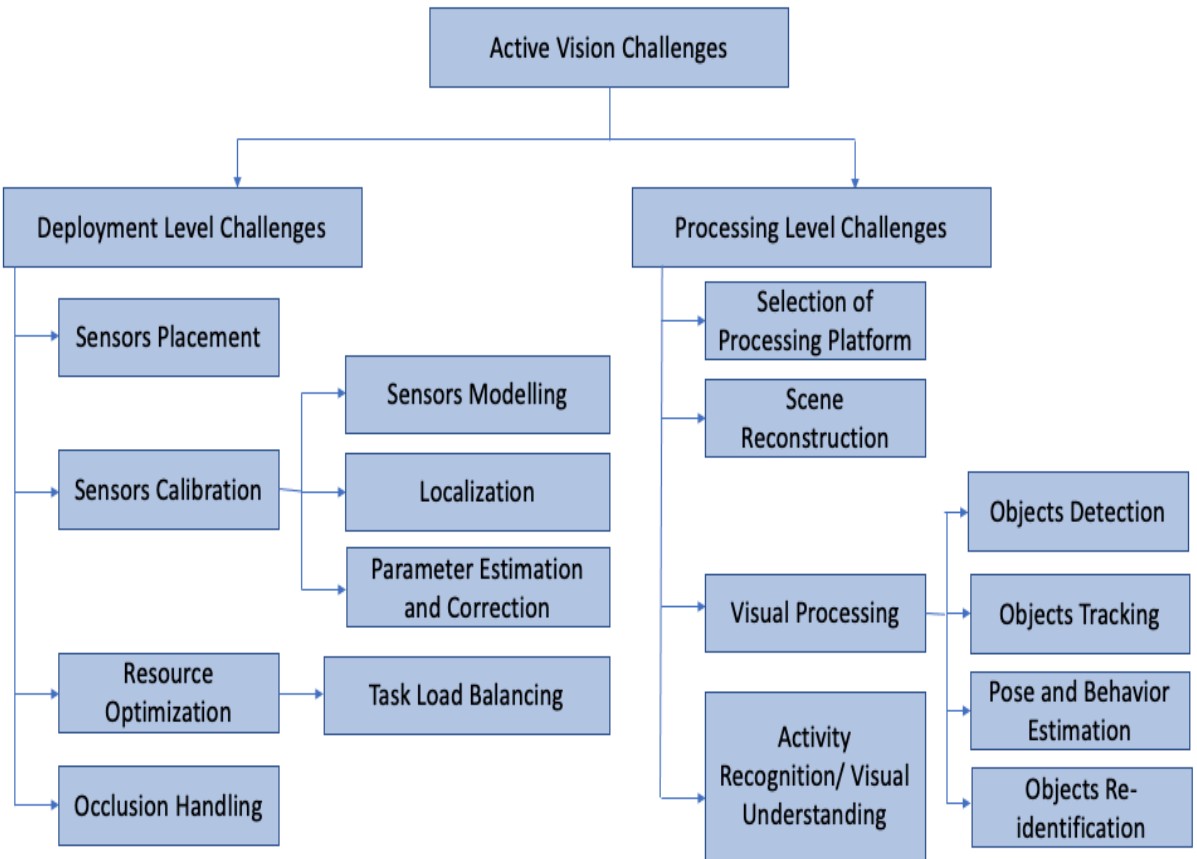

**Figure 2.** Taxonomy of challenges in active vision applications employing camera networks.

## 2. Challenges

A smart camera network (SCN) enhances the availability of raw data and thus the chances of deriving better information; however, it also increases the computational complexity of the active vision system of which it is a part. The configuration space of each sensor depends on the intrinsic parameters (such as optical center, focal length, etc.) and the extrinsic parameters of the sensor (such as rotation and translation, etc.), which require calibration [8].

Apparently, an active vision system utilizing a single fixed camera has much lower computational complexity and thus faces fewer challenges at the deployment level than an active vision system employing a network of mobile pan–tilt–zoom (PTZ) cameras. However, an active vision system employing multiple camera sensors can have an occlusion-avoidance capability that may be lacking in an active vision system employing a single camera for sensing.

### 2.1. Deployment-Level Challenges

Due to limited resources with respect to most of the sensor nodes and the high computational complexity of active vision systems, it becomes challenging to allocate tasks to the nodes. Additionally, optimization of the resources by each sensor node is equally important. Further, designing a sensor architecture and accurate sensor placement according to the overall objective of the system is another challenge at the deployment level. Furthermore, calibration of the sensors of the SCN and occlusion handling by the system also add to the challenges at the deployment level.

#### 2.1.1. Sensor Placement

Generally, cameras in SCNs are placed with overlapping FOVs to reproduce the entire operating environment. However, in some cases, with limited resources and larger operational areas, it becomes very difficult to place cameras with overlapping FOVs. The

placement of cameras has a direct impact on the quality and quantity of data available for processing. For example, if an object is captured in the center of a camera's FOV, the quality of the data, and thus the visual information available from the data, is much higher than if the object is captured at the edges of the camera's FOV. In addition, camera placement must ensure maximum coverage of events and thus camera placement becomes a critical part of the deployment of SCNs. For the above reasons, sensor placement is critical to the deployment of an SCN.

### 2.1.2. Calibration

The configuration space of an SCN includes the internal and external camera parameters of the sensor nodes or cameras of the SCN. It is critical to change the configuration space of each camera, considering the available and used resources. When the target changes its position in the field of view (FOV) of a sensor or moves from the FOV of one sensor to another, the configuration of the sensor must be changed by modifying its parameters so that data can be acquired to yield optimal information (i.e., by keeping the object as close to the center of the FOV as possible).

Some SCN designs also emphasize minimizing resource utilization by inactivating sensors if no activity is detected over a specified period of time, which alters the dynamic topology of the network. Thus, dealing with such dynamically changing situations is very challenging. At the operational level, calibration challenges are further classified as follows.

- *Sensor modelling*

The parameters of the configuration space of each sensor depend on the type of sensors used in the SCN. The sensor model provides information about the configuration space as well as the available resources, such as power, bandwidth, and the overall quality of service (QoS), possessed by the sensors. Therefore, sensor modelling plays a crucial role in active vision systems.

- *Localization*

When an object moves away from the FOV of a sensor, it is important to hand over the object to another sensor in the SCN. Since the topology of the SCN is dynamic, the network must be configured to receive information about the active nodes and their relative positions at all times. This information is obtained through the active localization of sensor nodes in the SCN. Localization plays an important role in the system's deployment, as it aids in determining the dynamic relative positions of active sensor nodes in the SCN.

- *Parameter estimation and correction*

Real-time calibration of the active sensors participating in the SCN is required to capture the objects of interest (OOIs) with maximum information. Therefore, parameter estimation and correction play important roles in active vision systems employing SCNs.

### 2.1.3. Resource Optimization and Task-Load Balancing

Most of the present-day active vision systems require sensor mobility and thus rely on batteries and wireless communication. Due to limited resources, optimum resource utilization becomes critical. Resource optimization can be achieved in two ways, either by using low-power-consuming components at the nodes or by using the sensor nodes smartly. Low-power-consuming components are generally expensive and thus add to the overall expense of deploying an SCN. In a pre-established network, employing new low-power-consuming components may lead to having to discard the entire system and build a new one from the scratch. On the other hand, using the sensors smartly (only when required) does not necessarily require changing the existing hardware or components in the SCN. However, getting a machine to learn when to activate while responding in real time can be challenging for a system. An outcome of a dynamically changing state of the nodes in the SCN is dynamic task-load allocation to the active nodes in the SCN.

The network topology changes dynamically as the sensors switch from active to inactive states. Additionally, in order to avoid deviating from the SCN's main goal (i.e.,

obtaining sensor data with high-quality visual information), the system's overall function­ality is dynamically distributed amongst the nodes in the form of task loads. As a result, the SCN must compute the dynamic topology in real time, determine the nodes' localizations, and simultaneously distribute the task load among the active nodes in the SCN.

### 2.1.4. Handling Occlusions

The OOI can occasionally become obscured while being observed by a sensor, resulting in low-quality data due to the occlusion blocking the OOI. One way to deal with this situation is to hand over the OOI to the subsequent nearest sensor node. However, finding a next-best node for the OOI in real-time also creates challenges. Moreover, it becomes difficult if any nearby nodes are unavailable or if the OOI is outside the field of view (FOV) of any nearby nodes. There are a few prediction-based approaches that rely on computation; however, these approaches lack accuracy and may result in missing critical data, making them unreliable.

### 2.2. Challenges in Data Processing

Processing sensor data to obtain understanding is the most critical part of any active vision system. Active vision systems utilize data from multiple sources (i.e., multiple sensor nodes) and therefore require highly complex computational capabilities. The visual processing becomes more challenging when an application has to deliver results in real time. The data-processing challenges of an active vision system are discussed below.

### 2.2.1. Selection of a Processing Platform

The choice of processing platform is as important as the algorithms used to process the sensor data. The processing requirements of active vision are selected based on the system's overall functionality, which is typically determined by two factors: processing time and functional complexity. For instance, the architecture of a system that requires complex computations in real time might be more complex and therefore more expensive than one with a relaxed processing window for applications with simpler computations.

### 2.2.2. Scene Reconstruction

The data obtained by a number of active sensors need to be synchronized to obtain useful information. An individual sensor's data need to be stitched together in such a way that they result in the determination of an action performed by the OOI. The dynamically changing network topology makes this even more challenging.

### 2.2.3. Data-Processing Challenges

The active vision system relies on visual-processing algorithms to achieve the overall functionality of the system. Data-processing challenges include detecting an event or activity by the OOI and further deriving an understanding about the detected event or activity. Various visual-processing challenges in an active vision system are discussed below.

- *Object detection*

While detecting an OOI, an active vision system divides the acquired sensor data into a foreground and a background. It should be noted that in an active vision system, object detection and data extraction by sensors are interdependent, making it difficult for any active vision system to accurately detect the OOI. In addition, one or more objects in the observed scene may be performing multiple tasks, increasing the likelihood of problems. Variations in viewpoints, shifts in lighting conditions, occlusions, and similar factors often lead to issues with object detection.

- *Object classification and tracking*

Detected objects undergo classification to distinguish one object from another. Object classification requires the selection of appropriate methods for pattern recognition, cluster­ing, and data segmentation. There are several methods for pattern recognition, clustering,

and segmentation and there are associated advantages and challenges. Therefore, the selection of an appropriate method for object classification is very critical. Further, information about the activity performed by the object(s) participating in an event under observation is extracted by tracking the OOI(s) in consecutive frames. Deciding tracking features based on the activity of an OOI can be challenging because the OOI perceived by the sensor in successive frames is likely to have different viewpoints (i.e., the shape of the OOI perceived by the sensors may appear variable).

- *Object Re-identification*

    Many active-vision applications rely on relating activities of an OOI observed in non-consecutive frames. For instance, an understanding of an event can be obtained by comparing the actions of an OOI seen in various camera FOVs and at various times. For the system to reach a conclusion, it must re-identify an already identified OOI and correlate different activities of the OOI performed at different times.

- *Pose and behavior estimation*

    Consecutive frames are subjected to spatio-temporal evaluations of consecutive frames to obtain pattern and pose information from the OOI tracked in the scene. An accurate relationship between the activities and the changes in the pose of the OOI is required to relate the visual information to the corresponding understanding of the activity. Further, the active vision system must be configured to estimate variation in the pose of the OOI and thus derive the behavior of the OOI based on the pattern of the poses.

### 2.2.4. Activity Recognition and Understanding

Activity detection and recognition follows two paradigms: static and dynamic. An activity can be detected by analyzing a single frame, known as static recognition, and requires only spatial evaluation of the frame. Dynamic approaches require the evaluation of multiple consecutive frames, utilizing scene reconstruction (also known as dynamic recognition), and thus require spatio-temporal computational capabilities. Static recognition processes do not require pose estimation. Dynamic recognition, on the other hand, is generally used to solve complex problems where the information is obtained by pose estimation, such that the pose information is subsequently processed.

It is worth noting that the process flow of an active vision system utilizing an SCN includes two basic components: one derived from the functionality of the SCN (i.e., for the dynamic calibration of parameters) and the other derived from the computer vision system (i.e., for visual processing). In addition, it must also be noted that both the abovementioned components are inter-related co-operatively.

### 2.3. System-Level Challenges

In an active vision system employing an SCN to extract sensor data, the calibration of parameters of the sensor nodes participating in the SCN is highly dependent on the detection and interpretation of the sensor data by the active vision system. Further, the effectiveness and performance of the active vision system (i.e., the data processing) depend on the quality of data extracted by the SCN nodes. Thus, it can be inferred that the functionality of the SCN and the data processing system are interdependent. Moreover, real-time decision making is even more challenging for such an interdependent system.

In some respects, an active vision system can be considered a type of computer vision system wherein the system is configured to make alterations in the configuration space of the sensor(s) employed by the system in real time in a manner pre-defined by way of design protocols. However, an active vision system is not capable of handling most unforeseen challenges because it is protocol-driven and therefore cannot make independent decisions. Thus, the system-level challenges mentioned above are very critical and hard to overcome. If the system is configured with self-reconfigurable properties, it can overcome the abovementioned challenges. If such a system is configured with adaptive capabilities, this may further

facilitate an enhancement in performance through active learning. The self-reconfiguration and adaptive properties of an active vision system are discussed later in Section 5.

## 3. Existing Solutions

To understand the development and progress at each operational level of active vision systems, we have performed an extended search survey for the last two decades. The results were generated with the keywords "active vision", "computer vision", "smart camera network", AND "the name of challenge", as discussed in Section 2, using the Google Scholar search tool.

### 3.1. Sensor Placement

The two major challenges to be addressed while deploying an SCN in reference to sensor placement are the maximization of the surveillance area and the handling of non-overlapping FOVs (i.e., managing handover). Indu et al. [10] and Zhang et al. [11] proposed methods for sensor placement in networks aiming to maximize the surveillance areas covered by the sensors. Silva et al. [12] proposed a system for coordination among the sensor nodes of an unmanned aerial vehicle (UAV) network for efficient surveillance. The systems and methods proposed in [10–12] present camera-placement solutions for optimized functionality; however, they lack flexibility in architecture and prioritization of surveillance space. An activity-based prioritization of the surveillance area is proposed in [13] by Jamshed et al., addressing the abovementioned concern. Vejdanparast [14] addressed the camera-placement problem for maximizing the area of surveillance by enhancing the fidelity of each camera in the network. Wang et al. [15] proposed Latin-Hypercube-based Resampling Particle Swarm Optimization (LH-RPSO) based on a camera placement algorithm for IoT devices and networks.

Redding et al. [16] proposed object handover based on multiple features, such as Zernike moments, scale-invariant feature transformation, gray-level co-occurrence matrices, color models, etc., using cross-matching for non-overlapping FOVs. In [17], Esterele et al. presented a method for the generation of an online real-time vision graph for the handover of information in a decentralized network with non-overlapping FOVs. The method proposed in [17] demonstrated that no prior knowledge of nodes is required and that it is easy to add or remove nodes from the network. Lin et al. [18] proposed an active handover control for real-time handover of single objects using multiple PTZ cameras, using the shortest distance rule and spatial relations. The method proposed in [18] proposed the readiness of a receiving camera before handover. A year-wise representation of the abovementioned sensor placement techniques along with their advantages is shown in Table 1.

**Table 1.** Evolution of sensor-placement techniques.

| Ref. | Year | Methodology | Advantages |
|------|------|-------------|------------|
| [16] | 2008 | Online system for tracking multiple people in an SCN with overlapping and non-overlapping views | Development of a larger, more capable, and fully automatic system without prior localization information |
| [10] | 2009 | Genetic algorithm | Maximum coverage of users; Defined priority areas with optimum values of parameters; The proposed algorithm works offline and does not require camera calibration; Minimizes the probability of occlusion due to randomly moving objects |
| [17] | 2011 | Ant-colony-inspired mechanism used to grow the vision graph during runtime | Generates a vision graph online; Increased autonomy, robustness, and flexibility in smart camera networks |

**Table 1.** *Cont.*

| Ref. | Year | Methodology | Advantages |
|------|------|-------------|------------|
| [18] | 2012 | Approach to construct the automatic co-operative handover of multiple cameras for real-time tracking | Tracking a moving target quickly and keeping the target within the viewing scope at all times |
| [11] | 2015 | Novel model with non-uniformly distributed detection capability (DC) | Orientation of each visual sensor can be optimized through a least-squares problem; More efficient with an averaged relative error of about 3.4% |
| [13] | 2015 | Node-level optimal real-time priority-based dynamic scheduling algorithm | Portable system with ease of access in hard-to-access areas |
| [12] | 2017 | Coordination of embedded agents using spatial coordination on strategical positioning and role exchange | Persistent surveillance with dynamic priorities |
| [14] | 2020 | Novel decomposition method with an intermediate point of representation | Low computational expense; Higher fidelity of the outcomes |
| [15] | 2020 | Latin-Hypercube-based Resampling Particle Swarm Optimization (LH-RPSO) | LH-RPSO has higher performance than the PSO and the RPSO; LH-RPSO is more stable and has a higher probability of obtaining the optimal solution |

### *3.2. Calibration*

The major challenges in the calibration of nodes participating in an SCN concern sensor (camera) modelling, localization, and parameter estimation and correction. Some of the proposed solutions addressing various calibration challenges are discussed hereinbelow.

### 3.2.1. Camera Modelling

Some basic models for camera calibration are the thin-lens camera model, the pinhole camera model (linear-perspective projection model), the orthographic projection model, the scaled orthographic projection model, and the para-perspective projection model.

The thin-lens model is a linear calibration model that accounts for the effects of translation and rotation relative to a view plane. The pinhole model later introduced the effect of linear perspective projection; however, it has high computational complexity.

To overcome the high computational complexity of the pinhole model, Hall et al. [19] proposed a much simpler and computationally efficient linear model based on 3D affine transformation with linear perspective projection. The abovementioned linear models did not perform well, as they were unable to account for non-linear distortion, which was addressed by improving 3D affine transformation with non-linear perspective projection models by Tsai et al. in [20], Toscani's non-linear calibration model in [21], and by Wang et al. in [22].

### 3.2.2. Localization

Camera localization helps estimate the relative positions, orientations, and poses of active nodes in a network. Identifiers or markers in the form of lines, points, features, cones, circles, spheres, etc., are commonly used for the localization of nodes in a camera network.

Such identifiers are commonly used in unknown network environments but can also be used in known environment for improved accuracy and better scene mapping. Utilizing perspective points as markers, the Perspective-n-Point (PnP) algorithm can be used in a known network environment. Simultaneous localization and mapping (SLAM), as presented in [23,24], and structure from motion (SFM), as presented in [25], can be used for dynamically changing network environments. The SFM technique in [25] is based on human vision perseverance to estimate a 3D scene using 2D image data by combining image motion information with frame data. The Monte Carlo method in [26] uses a particle filter for localization and recursive Bayesian estimation for sorting and sampling.

Montzel et al. [27] proposed a distributed energy-efficient camera network localization method using sparse overlapping in 2004. In [28], Brachmann and Rother proposed 6D

pose estimation using an end-to-end localization pipeline. The geometric localization obtained via the head-to-foot location (poles) of pedestrians using an estimated distribution algorithm (EDA) in [29] can be utilized for self-calibration of the nodes in a network. A year-wise representation of the abovementioned localization techniques along with their advantages is shown in Table 2.

**Table 2.** Evolution of localization techniques.

| Ref. | Year | Methodology | Advantages |
|:---:|:---:|:---:|:---:|
| [26] | 1999 | Online system for tracking multiple people in an SCN with overlapping and non-overlapping views | Development of a larger, more capable, and fully automatic system without prior localization information |
| [27] | 2004 | Sparse overlapping | Better energy efficiency and able to cope with networking dynamics |
| [23] | 2006 | SLAM | Locally optimal maps with computational complexity independent of the size of the map |
| [24] | 2006 | SLAM | Locally optimal maps with computational complexity independent of the size of the map |
| [29] | 2016 | Estimated distribution algorithm (EDA) | Accurate estimation of the features of moving objects (person) |
| [25] | 2017 | SFM | Better ambiguity handling in 3D environments |
| [28] | 2018 | 6D pose estimation using an end-to-end localization pipeline | Efficient, highly accurate, robust in training, and exhibits outstanding generalization capabilities |

### 3.2.3. Parameter Estimation and Correction

Zheng et al. [30] proposed a focal-length estimation method using parallel particle swarm optimization (PSO) with low time complexity and efficient performance. Führ and Jung [31] proposed a self-calibration method for the surveillance of pedestrians in a static camera network using a projection matrix obtained from non-linear optimization of an initial projection matrix obtained after pole extraction. The information in the projection matrix was used for the localization of cameras in the network. In [32], Yao et al. proposed a self-calibration model for dynamic multi-view cameras using golf and soccer datasets based on a field model.

Li et al. [33] proposed a greedy-descent-optimization-based parameter-estimation and scene-reconstruction framework for camera–projector pairs for self-calibration. A network of such systems can be used for efficient tele-immersion applications. Janne and Heikkilä [34] proposed a self-reconfiguration solution for a camera network with focal-length estimation using homography from unknown planar scenes. In [35], Tang et al. proposed a simultaneous distortion-correction self-configuration method using an evolutionary optimization scheme on an estimated distribution algorithm (EDA) for tracking and segmentation. A year-wise representation of the abovementioned parameter estimation techniques along with their advantages is shown in Table 3.

### 3.3. Resource Optimization: Topology Estimation and Task-Load Balancing

As the network topology changes, the task load needs to be altered to ensure that the overall functionality of the system is achieved. Marinakis and Dudek [36] proposed a system to estimate the topology of a visual network in the form of a weighted directed graph using statistical Monte Carlo expectation and sampling models. Hangel et al. in [37] addressed the problem of topology estimation for a large camera network and proposed a window-occupancy-based method as a solution. The method in [37] required a lot of assumptions and could not handle large numbers of data. Detmold et al. [38] proposed a topology-estimation method capable of handling data from a large number of nodes in the network by scalable collective stream processing using an exclusion algorithm in distributed clusters of nodes. The method proposed by Detmold et al. [38] is similar to the

decentralized processing scheme. Clarot et al. [39] proposed an activity-matching-based network topology for distributed networks. Zhou et al. [40] proposed topology estimation by means of a statistical approach in a distributed network environment, utilizing identity and appearance similarity.

**Table 3.** Parameter-estimation techniques.

| Ref. | Year | Methodology | Advantages |
|------|------|-------------|------------|
| [31] | 2015 | Projection matrix obtained from non-linear optimization | Better accuracy |
| [32] | 2016 | Field model | Automatic estimation of camera parameters with high accuracy |
| [33] | 2017 | Greedy descent optimization | Stable and robust automatic geometric projector camera calibration with high accuracy; Efficient in tele-immersion applications |
| [34] | 2017 | Homography from unknown planar scenes | Highly stable |
| [30] | 2018 | Parallel particle swarm optimization (PSO) | Low time complexity and efficient performance |
| [35] | 2019 | Evolutionary optimization scheme on an EDA | Capability of reliably converting 2D object tracking into 3D space |

In [41], Farrel and Davis proposed network topology estimation for decentralized data processing. Zhu et al. [42] proposed a centralized processing approach for topology discovery using pipeline processing of lightning variations. In [43], Goutam and Misra proposed a trust-based topology management system for a distributed camera network. Tan et al. [44] proposed a method for topology estimation using blind distance as a parameter. In [45], Li et al. proposed topology estimation using Gaussian and mean cross-correlation functions for a distributed camera network. A year-wise representation of the abovementioned topology-estimation techniques along with their advantages is shown in Table 4.

**Table 4.** Topology-estimation techniques.

| Ref. | Year | Methodology | Advantages |
|------|------|-------------|------------|
| [36] | 2005 | Monte Carlo expectation maximization and sampling | Minimum effects of noise and delay |
| [37] | 2006 | Window-occupancy-based method | Efficient and effective way to learn an activity topology for a large network of cameras with a limited number of data |
| [38] | 2007 | Exclusion algorithm in distributed clusters | High scalability |
| [40] | 2007 | Statistical approach in distributed network environment | Robustness with respect to appearance changes and better estimation in a time varying network |
| [41] | 2008 | Decentralized data processing | Robustness with respect to variable appearance and better scalability |
| [39] | 2009 | Activity-based multi-camera matching procedure | Flexible and scalable |
| [42] | 2015 | Pipeline processing of lightning variations | Automated tracking and re-identification across large camera networks |
| [43] | 2015 | Trust-based topology management system | Higher average coverage ratio and average packet delivery ratio |
| [44] | 2018 | Blind-area distance estimation | Finer granularity and high accuracy |
| [45] | 2018 | Gaussian and mean cross-correlations | Better target tracking under a single region and better interference in multi-view regions |

An efficient computer vision system differentiates the overall functionality into a number of small tasks to optimize the system's functionality. The task load for each active

node depends on its local state, orientation, and available resources. In some respects, resource utilization is related to task-load balancing at each active node in the SCN. Kansal et al. [46] presented a distributed approach for adaptive task-load assignment on the basis of available energy from the network environment, which significantly improved the lifetime of the system. Rinner et al. [47] proposed a heterogeneous multiple mobile-agent-based task-allocation framework utilizing a distributed multi-view camera network. Later, Rinner et al. [48] presented an updated approach to allocate tasks for traffic surveillance, proposing clustered surveillance areas. In [49], Karuppiah et al. proposed a hierarchy-based automatic resource allotment and task-load balancing algorithm using fault tolerance based on activity density for a distributed network. Dieber et al. [50] proposed expectation-maximization-based task-load assignment to optimize monitoring performance, with efficient resource utilization. Dieber et al. [51] extended their work on task-load balancing with market-based handover for real-time tracking with optimized resource utilization. In [52], Christos et al. proposed a market-based bidding framework for multi-task allocation for a distributed camera configuration. A year-wise representation of the abovementioned task-load balancing techniques along with their advantages is shown in Table 5.

**Table 5.** Task-load balancing techniques.

| Ref. | Year | Methodology | Advantages |
|:---:|:---:|:---:|:---:|
| [46] | 2003 | Method for distributed adaptive task-load assignment | Better resource efficiency |
| [47] | 2005 | Multiple-mobile-agent-based task-allocation framework | Selective operation of the tracking algorithm to reduce the resource utilization |
| [48] | 2005 | Multiple-mobile-agent-based task-allocation framework | Selective operation of the tracking algorithm to reduce the resource utilization |
| [49] | 2010 | Hierarchy-based automatic resource allotment | Robust tracking |
| [50] | 2011 | Expectation-maximization-based approximation | Efficient approximation method for optimizing the coverage and resource allocation |
| [51] | 2012 | Market-based handover | Improved quality of surveillance with optimized resources |
| [52] | 2016 | Market-based handover | Improved quality of surveillance with optimized resources |

*3.4. Occlusion Handling*

Occlusion-handling approaches either aim at handing over the OOI to the next-best sensor node or predicting the occluded part of the OOI and reproducing it virtually to obtain the missing visual information. Occluded objects in the camera field can result in loss of activity information and thus compromise the functionality of the control system utilizing the understanding provided by the computer vision. Wang et al. [53] proposed occlusion estimation at each point of a scene flow field with patch-match optimization utilizing feature consistency and smoothness regularization as performance parameters in space with an improved red–green–blue dense model. In [54], Quyang et al. proposed a framework based on a part-based deep model for pedestrian detection. The proposed model is capable of estimating information loss due to occlusion in the form of errors in detector scores, using visibility of parts as a parameter.

Shahzad et al. [55] proposed multi-object tracking with effective occlusion handling by modelling the foreground using a K-means algorithm, where the object information is associated after occlusion using a statistical approach. Rehman et al. [56] proposed clustering based on a variational Bayesian method and multi-object tracking based on concepts of attractive and repulsive forces depending upon Euclidean distances between objects, utilizing a social force model to avoid the effects of occlusion. In [57], Chang et al. proposed a convolutional-neural-network-(CNN)-based tracking system with sparse coding for the pre-training of a network with the capability of handling occlusion effectively

for the surveillance and classification of vehicles. Zhao et al. [58] proposed an adaptive background formulation based on a Gaussian model for occlusion handling and object tracking in a coarse-to-fine-manner without-affecting-appearance model present in the system. In [59], Liu et al. proposed a distraction-aware tracking system based on a 3D mean-shift algorithm, capable of altering its appearance model and occlusion handling by utilizing depth information of the OOI. A year-wise representation of the abovementioned occlusion handling techniques along with their advantages is shown in Table 6.

**Table 6.** Occlusion-handling techniques.

| Ref. | Year | Methodology | Advantages |
|------|------|-------------|------------|
| [53] | 2015 | Patch-match optimization | Reduced computational complexity by large displacement motion |
| [54] | 2015 | Part-based deep model | Handles illumination changes, appearance change, abnormal deformation, and occlusions effectively |
| [56] | 2015 | Social force model | Improved tracking performance in the presence of complex occlusions |
| [55] | 2016 | K-means algorithm and statistical approach | Cost-effective in terms of resources (memory and computation) |
| [58] | 2017 | Gaussian model for occlusion handling | Handles appearance changes and is capable of dealing with complex occlusions |
| [57] | 2018 | CNN | High performance with a limited labelled training dataset |
| [59] | 2018 | Distraction-aware tracking system | Effective and computationally efficient occlusion handling |

### 3.5. Selection of a Processing Platform

Selection of a platform to develop a computer vision system is as critical as designing or selecting algorithms specific to the functionality. Most commonly, the processing platforms may either be software-based, as with a central processing unit (CPU) or a graphical processing unit (GPU), or hardware-based, as with a field-programmable gate array (FPGA) and application-specific integrated circuits (ASICs). The selection of the platform depends on the requirements of processing capabilities, result accuracy, flexibility, timeliness, and resource utilization. A comparative evaluation of the selection of processing platforms for computer vision systems is presented by Feng et al. [60].

Systems that require flexible functionality usually prefer CPU- or GPU-based processing platforms; however, the efficiency of such systems is low. On the other hand, ASICs and FPGA are used for systems that require high efficiency and better and faster computations; however, these systems lack operational flexibility.

Hørup et al. [61] presented a comparative analysis of general-purpose computations performed by CPUs and GPUs in computer vision systems. Guo et al. [62] proposed a fast and flexible CPU-based computation system for human pose estimation. Tan et al. [63] proposed a fast yet flexible deep-learning-based computer vision system utilizing a GPU. Irmak et al. [64], Costa et al. [65], and Carbajal et al. [66] proposed FPGA-based computer vision systems, whereas Xiong et al. [67] presented an ASIC-based computer vision system with enhanced operational flexibility. To obtain the advantages of the two kinds of platform, a hybrid model, i.e., a platform with a hardware–software combination such as the one presented in [68], can also be utilized for computation.

The state-of-the-art research on the selection of processing platforms aims to serve the ends of efficiency and flexibility of computation. CPU- and GPU-based systems aim to improve efficiency and computational speed, whereas systems based on FPGA and ASICs are intended to enhance operational flexibility.

### 3.6. Scene Reconstruction

The sensors in the SCN obtain raw sensor data from their respective FOVs. The raw data are then fused together, utilizing spatio-temporal information (obtained through the frame count and relative localization of sensors) associated with the data. The integration

of sensor data into a virtual environment or a scene is called scene reconstruction. R. Szeliski [69] proposed a novel volumetric scene-reconstruction method using a layered structure and multiple depth maps. Martinec et al. [70] proposed 3D reconstruction using an uncalibrated image dataset and a pipelining approach to detect regions of interest (ROIs) and match them using random sample consensus (RANSAC). Peng et al. [71] addressed the network geometry-estimation problem utilized for scene reconstruction and proposed two-view geometry estimation using a local-structure-constraint-based L2-estimation–local-structure-constraint (L2E-LSC) algorithm.

For efficient scene reconstruction, effective point matching is imperative. Brito et al. [72] compared different state-of-the-art point correspondence methods, such as Scale Invariant Feature Transform (SIFT), Fast Retina Keypoints (FREAK), Oriented Fast and Rotated Brief (ORB), Binary Robust Invariant Scalable Keypoints (BRISK), and Speeded-Up Robust Features (SURF). Milani [73] introduced localization-based reconstruction for a heterogeneous network. Aliakbarpour et al. [74] reviewed different scene-reconstruction methodologies and presented a reconstruction method using parametric homography. Wang and Guo [75] presented a reconstruction method using plane primitives of an RGB-D frame. Ma et al. [76] proposed a mesh-reconstruction system using an adaptive octree division algorithm for point-cloud segmentation and mesh relabeling and reconstruction for scene reconstruction. Ichimru et al. [77] presented 3D scene reconstruction using a CNN under water, utilizing transfer learning with a bubble dataset to avoid distortions.

### 3.7. Data Processing

The overall objective of a computer vision system is to detect an activity or event of interest, perform processing on the data containing information about the event, develop an understanding of the event, and generate an action by way of a control unit based on the understanding. The visual processing includes a number of stages. Recent advancements in the most common stages of visual processing in computer vision systems are discussed hereinbelow.

### 3.7.1. Object Detection

The first step towards the visual processing of data in a computer vision system is detection of the OOI (also referred to as the foreground). Some traditional object-detection methods are the Viola and Jones technique [78], scale-invariant feature transformation (SIFT) [79], HOG-based detection [80], optical flow [81,82], and background subtraction [83]. Most of the recent computer vision systems involve machine-learning-based object detection, including neural-network-based object detection [84], "you only look once" (YOLO) [85], region proposals (R-CNN) [86], single-shot refinement neural networks (RefineDet) [87], Retina-Net [88], and single-shot multi-box detectors (SSDs) [89]. The recent neural-network-based object-detection methods provide much better accuracy as compared to the traditional detection methods, but they are highly dependent on the training data. A survey of the evolution of detection techniques in computer vision from probabilistic-prediction approaches to advanced machine-learning approaches is presented in [2].

Some challenges in object detection arise due to dynamic illumination changes, the movement of objects, and occlusions. Roy and Ghosh [90] proposed an adaptive background model (a histogram min–max bucket) using a single sliding window to add adaptability to the background detection. The adaptive background model used a median-finding algorithm incorporated to handle dynamic illumination changes. Bharti et al. [5] proposed an adaptive real-time occlusion-handling kernelized correlation framework for UAVs capable of updating location and boundary information based on the confidence values of the tracker. Min et al. [91] proposed a multiple-object-detection approach using pixel lifespan to blend ghost shadows to the background and a classifier based on a state vector machine (SVM) and a convolutional neural network (CNN) to avoid occlusions.

Detecting an object in a moving FOV is a tedious task and requires substantial approximations. Wu et al. [92] proposed an effective computational model to solve this problem. They evaluated a coarse foreground using singular-value decomposition and

reconstructed the background using the foreground information obtained through a fast in-painting technique. Mean-shift segmentation was used for further refinement of the foreground. Hu et al. [93] presented a tensor-based approach to detect mobile objects without changing the scene dynamics. For initial foreground detection, saliently fused sparse regularization was used and tensor nuclear norms were utilized to handle background redundancy. The foreground was further improved using a 3D locally adaptive regression kernel, which was used to compute spatio-temporal variations. A year-wise representation of the abovementioned object detection techniques and their advantages is shown in Table 7.

**Table 7.** Object-detection techniques.

| Ref. | Year | Methodology | Advantages |
|------|------|-------------|------------|
| [83] | 1989 | Background subtraction | Low computational complexity |
| [78] | 2001 | Viola and Jones technique | Low processing latency with high detection rate |
| [80] | 2005 | HOG-based detection | Precise object detection and classification |
| [79] | 2012 | Scale-invariant feature transformation | Efficient detection and localization of duplicate objects under extreme occlusion |
| [81] | 2013 | Optical flow | Accurate detection of moving objects |
| [86] | 2014 | Region proposals (R-CNNs) | High accuracy and precision for object detection |
| [92] | 2015 | Background subtraction and mean shift | Refined and precise foreground detection |
| [85] | 2016 | "You only look once" | Low latency multi-object detection |
| [89] | 2016 | Deep-neural-network-based SSD | Prediction-based detection for variable shapes of objects |
| [93] | 2016 | Tensor flow | Detection of mobile objects in FOVs |
| [84] | 2017 | Neural network | Multi-object detection with variable shapes |
| [90] | 2017 | Adaptive background subtraction model | Better accuracy as compared to traditional background subtraction |
| [91] | 2017 | State-vector machine and CNN-based classifier | Multiple-object-detection approach to detect ghost shadows and avoid occlusions |
| [82] | 2018 | Optical flow | Accurate detection of moving objects |
| [87] | 2018 | Single-shot refinement neural network | High detection accuracy |
| [5] | 2018 | Kernelized correlation framework | Real-time occlusion handling |
| [88] | 2019 | Retina-Net | Balanced detection performance in terms of latency, accuracy, and precision of detection |

### 3.7.2. Object Classification and Tracking

The OOI can be classified after detection on the basis of one or more appearance parameters [94] called features that classify the object based on color, texture, shape, pixel motion, etc. Some basic features for object representation presented in [95] are points, shapes, and silhouettes or contours. Conventionally, object-classification methods [96] can be categorized as decision-based, statistical-probability-based, and soft-computing-based techniques. Some common decision-based classification methods are decision trees [97,98] and random forests [99]. Bayesian classification [100–102], discriminant analysis [103], logical regression [104], and nearest-neighbor [105] approaches use statistical probability for classification. State vector machines [106], multi-layered perceptrons [107], and neural networks [108,109] use soft computing for classification.

One of the major challenges in object tracking is distortion of the OOI. Villiers et al. [110] proposed real-time inverse distortion for distortion correction. A lot of methods for distortion correction and calibration use properties of vanishing points, as was first proposed in [111]. A distributed algorithm proposed by Caprile et al. [35] illustrated the utilization of tracking waking humans as poles to derive vanishing points for radial distortion

correction and self-calibration. Radial distortion correction was addressed in [112,113] by estimating the center of distortion. Huang et al. [114] proposed linear-transformation-based radial distortion correction, whereas Zhao et al. [115] used a pipelined process for radial distortion correction. Methods for the correction of radial as well as tangential distortion were proposed in [116–118]. Yang et al. [119] proposed estimation and correction of perspective distortion utilizing depth information. Color-calibration theory, discussed by Finlayson et al. in [120], has been used to address the challenge of optical distortion. In [121], Wong et al. presented a color-calibration approach using a multi-spectral camera model.

Another challenge in object tracking is to obtain the best possible information for each object of interest in the scene while dealing with multiple OOIs in the scene. One of the factors affecting the performance of this task is the motion of the camera nodes to capture the objects precisely. The motion blur sometimes increases to such an extent that the effect of moving the camera in the environment is nullified. Han et al. [122] proposed a motion-aware tracker to address the abovementioned problem by filling in the tracking fragments caused by occlusion or blur. Meinhardit et al. [123] proposed a former tracking system to address the challenges of multi-object tracking.

### 3.7.3. Object Re-Identification

Object re-identification encounters multiple challenges through occlusions, false object detection due to ghost shadows [124], illumination changes, and change in viewpoints. Zhang et al. [125] proposed an adaptive re-identification framework for spatio-temporal alignment utilizing Fisher vector learning to address illumination changes in the re-identification of OOIs. Yang et al. [126] used logical determinant metric learning to tackle re-identification through different camera views to overcome occlusions in the re-identification of objects. Multiple features, when fused together, improve re-identification capabilities; however, the role of each feature and the weights are critical. Geng et al. [127] proposed a feature-fusion method based on weighted-center graph theory to obtain the role of each feature in re-identification. Yang et al. [128] presented re-identification using partial information which can be utilized with occluded data.

### 3.7.4. Pose and Behavior Estimation

Pose-estimation methods utilize models to associate the patterns of postures, poses, or shapes of OOIs detected and track them to obtain meaningful information, thus yielding understanding. The major challenges in pose and behavior estimation are: the selection of a pose-estimation model and the association of information with a sequence of poses obtained through tracking an OOI. Pose-estimation methods can either be model-based and may use kinematic modelling, planar modelling, or volumetric modelling, or they can be model-free. Kinematic models rely on tracking the movements of points on the OOI, planar models rely on contours, whereas volumetric models rely on changes in the volume distributions of OOIs tracked over time. Kinematic models are easy to process but are not reliable. Planar and volumetric models are more reliable but have more computational complexity. Addressing the challenge of low computational complexity and high accuracy in pose and behavior estimation, Chen et al. [129] proposed an anatomically aware 3D pose-estimation model for human behavior analysis. Staraka et al. [130] proposed a kinematic skeletal-model-based pose-estimation method with real-time and accurate behavior estimation.

### 3.8. Visual Understanding

Systems achieve visual understanding by relating one or more pose behaviors and performing spatio-temporal analyses of the patterns of poses corresponding to events in a scene. Campbell et al. [131] utilized phase-space constraints to depict human motion. Oren et al. [132] used single-frame wavelet templates for pedestrian detection. Image captioning [133], manuscript reviewing in the medical field [134] and academia [135] are some applications of static activity recognition. Nguyen et al. [136] used multi-objective optimization for real-time activity monitoring. In [3,137], dynamic recognition was utilized

for sports analysis. Xiang et al. [138] proposed a system capable of making decisions for multiple-object tracking, whereas Wu et al. [4] proposed a dynamic activity recognition system for smart homes. In [139], Laptev et al. proposed a state-vector-machine-based abnormal human activity recognition system.

## 4. Contemporary Solutions

From the above, it can be observed that there has been a shift from model-based approaches to artificial-intelligence (AI)-based methods. We have also considered some of the earlier publications to understand the concepts of some of the recently proposed models. Further, it has also been observed that the emerging research fields in active vision systems are multi-object detection and tracking, occlusion handling, and sensor reconfiguration.

Artificial intelligence (AI) drives down the time taken to perform a task. It enables multi-tasking and eases workloads for existing resources. AI further facilitates decision making by making the process faster and smarter. For these reasons, most of the state-of-the-art systems addressing the abovementioned challenges rely on one or more artificial-intelligence (AI)-based approaches [140–156]. According to market research reported in [157], AI in computer vision has a 45% compound annual growth rate, which is the growth rate of research in the area. The main reason to switch from traditional model-driven approaches to artificial-intelligence-based systems is the high level of accuracy AI systems can provide as compared to the former. It has also been observed that most of the AI-based approaches are dependent on machine-learning (ML)- or deep-learning (DL)-based models.

Some of the recently proposed systems utilize the concepts of traditional model-based methods along with modern AI-based approaches to generate highly accurate hybrid systems [147,151], while some systems [153] capable of addressing the challenges through AI-based models utilize the basic principles of traditional model-based methods. Some of the state-of-the-art AI-based systems addressing the challenges of self-reconfiguration faced by active vision systems are presented in Table 8.

ML-based systems highly rely on training datasets to develop operational models. To make an ML-based system adaptive, AI models need to learn the unforeseen by deriving information from experience. Such systems fail drastically in centralized networks, as the nodes are trained on different datasets based on their specific experiences of distinct events in their surroundings.

Thus, the biggest challenge for adaptive systems is the sharing of information about events between the nodes in networks such that each event in the network of networks can be handled with equal accuracy by each node.

ML systems further suffer from visual attacks [158], the most common of which are adversarial attacks [159,160] which misguide the classifier and thus impact the accuracy of the system. One study [161] showcases the effects of adversarial attacks on the performance of machine-learning-based approaches. The systems are attacked by visual attacks which make small alterations in the weights of the classifier, thus misleading the classifier and reducing the accuracy of classification. Over a period of time, such attacks result in drastic degradation in the performance of the system. Refs. [162,163] present detailed surveys on the types of visual attacks and the methods proposed to detect and mitigate such attacks. Visual attacks [158] can be targeted if the effect of the attacks is predicted correctly by the model; however, in most scenarios, such attacks introduce random noise and thus are very difficult to reverse, causing permanent degradation in the classifier's performance over time. To overcome the problems of data loss and degradation of system performance due to such attacks, ML systems must be incorporated into a distributed network, such that the data are distributed throughout the entire network. Such a network is capable of detecting an attack in the initial stage itself and providing back-up for any kind of data loss to each participating node in the network.

**Table 8.** State-of-the-art AI-based approaches addressing reconfiguration and active-vision challenges.

| Ref. | Challenge Addressed | AI-Based Approach Used |
|---|---|---|
| [140] | Camera calibration | Convolutional neural network (CNN) |
| [141] | | Neural network |
| [142] | Parameter estimation | Convolutional neural network (CNN) |
| [143] | | Deep neural network (DNN) |
| [144] | Pose estimation | Neural network |
| [145] | Object detection | Modified CNN |
| [146] | Object tracking | Residual neural network |
| [147] | | Deep CNN and Kalman filter |
| [148] | | Deep neural network (DNN) |
| [149] | | CNN and deep sort |
| [150] | | Deep-learning-based CNN |
| [6] | Activity detection | Slow–fast CNN |
| [151] | | Neural network and strider algorithm |
| [152] | Object re-identification | CNN |
| [153] | | Sparse graph-wavelet-based CNN |
| [154] | Object re-identification and occlusion handling | Deep-neural-network-based transfer learning |
| [155] | Localization | CNN |
| [156] | | Neural network |

## 5. Self-Adaptation and Self-Reconfiguration

The challenges discussed above have received solutions designed to tackle particular problems in well-known settings. However, none of the aforementioned approaches addresses the challenge with unforeseen conditions optimally due to unexpected changes in the environment. To overcome this limitation, an active vision system must be capable of understanding the changes in its environment and reconfiguring the parameters of the active sensors participating in the system on its own. Further, active vision systems must also enable the sharing of information with each other to handle unforeseen changes in a better way.

An active vision system capable of self-adjusting its configuration space is called a "*self-reconfigurable*" system. The ability of an SCN to adapt to such changes and reconfigure its parameters for optimized performance in dynamic and unforeseen conditions is called self-reconfiguration of the SCN [164]. Such a self-reconfigurable SCN for UAVs used for surveillance was proposed by Leong et al. in [165]. A comparative survey presented by Natarajan et al. [166] illustrates a number of self-reconfiguration models for computer vision systems with multiple active nodes participating in data extraction. Martinel et al. [167] proposed the distributed self-reconfiguration of an SCN to address the vehicle re-identification problem utilizing deep-learning models.

The capability of a system to share information to learn and adapt can be achieved through "*self-adaptation*". Self-adaptation is the ability of a network (in this case, specifically an SCN) to enhance its performance by enabling the network to update the configuration space of nodes, active participants, protocols, and functional algorithms. Some systems configured for self-adaptation require learning from past experience or the sharing of performance statistics from different parts of the network and opt for the settings which worked best for similar events.

Self-adaptation can be achieved through self-expression and self-awareness capabilities in a system. A system having self-awareness (SA) possesses knowledge of its state;

self-awareness gives the system the ability to share its state in the form of parameters and overall quality of service (QoS). Additionally, each node in the self-adaptive system possesses the ability to question its current state, investigate alternative configurations for better QoS, and change its state using active learning.

Rinner et al. [168] used a market-based approach for self-awareness and proposed six major steps for self-adaptation, namely, resource monitoring, object tracking, topology learning, object handover, strategy selection and objective formation. Lewis et al. [169] classified events as explicit and implicit and discussed the privacy, extent, and quality of self-adaptation. Lewis and Chandra [170] discussed formal models for self-adaptation and the application of self-adaptation in systems with artificial intelligence, conceptual systems, engineering, automotive systems, computing, etc. Wang et al. [171] discussed methods of self-adaptation with capabilities of online learning. In [172], Ali et al. proposed an auto-adaptive multi-stream architecture using multiple heterogeneous sensors with pipelined switches between processing states and ideal states to reduce power, using an FPGA implementation that demonstrated inter-frame adaptation capability with a relatively low overhead. Guettalfi et al. [173] proposed an architecture utilizing quality of service (QoS), resource estimation, a feedback mechanism, and state estimation for public and private self-awareness using actuators. Zhu et al. [174] and Lin et al. [175] proposed a self-adaptation-based person-re-identification system based on unsupervised learning. Wu et al. [176] and Rudolph et al. [177] proposed an adaptive self-reconfiguration framework for computer vision systems. Both frameworks proposed the sharing of information between the sensors in a network for its utilization. However, both adaptive self-reconfiguration frameworks were designed for centralized network configurations and thus possess limited scopes of learning.

An adaptive self-reconfigurable framework presented hereinbelow, when utilized for an SCN-enabled active vision system, can provide adaptive calibration of SCN sensor parameters in near real time. Further, in the data-processing part of the system, the framework can be utilized to adapt to the best specifications by learning from the experiences obtained from other sources and thus result in the minimization of re-iterative training for models that are specific to the development of scene understanding.

## 6. Adaptive Self-Reconfiguration Framework

Piciarelli et al. [8] proposed a dynamic reconfiguration framework for SCNs, as shown in Figure 3. The framework used a local state (f), resources (r), and QoS information (q) from a number of nodes to generate the overall state (F), overall resources (R), and overall quality (Q) of the system, utilizing an SCN. The dynamic reconfiguration framework showcased the functionality of a reconfigurator, which was configured to determine changes in parameters based on a resource model and the objectives of the system. The dynamic reconfiguration framework in [8] was designed for an independent SCN system working in a centralized environment and thus lacks adaptiveness to deal with unforeseen events. The framework needs to reconfigure itself from scratch if a new type of activity is discovered and thus is not suitable for real-time active-vision applications. Further, almost all the symmetric/asymmetric systems and frameworks discussed in Sections 3 and 4 lack adaptiveness and thus are likely to fail miserably in unforeseen conditions. Recent reconfiguration systems [177–179] based on the reconfigurator model of [8] have tried to improve the accuracy of detection; however, due to centralized networks of operation, they have limited scopes for learning and are not suitable to tackle unforeseen conditions in real time.

As discussed earlier, the reconfiguration of sensor nodes is highly reliant on the scene understanding gained by the processing of sensor data and vice versa. Due to this interdependency, it is very challenging for a system to reconfigure its configuration space while dealing with unforeseen situations. With prior knowledge of an event, scene understanding can be greatly improved and thus the reconfiguration of the sensor network.

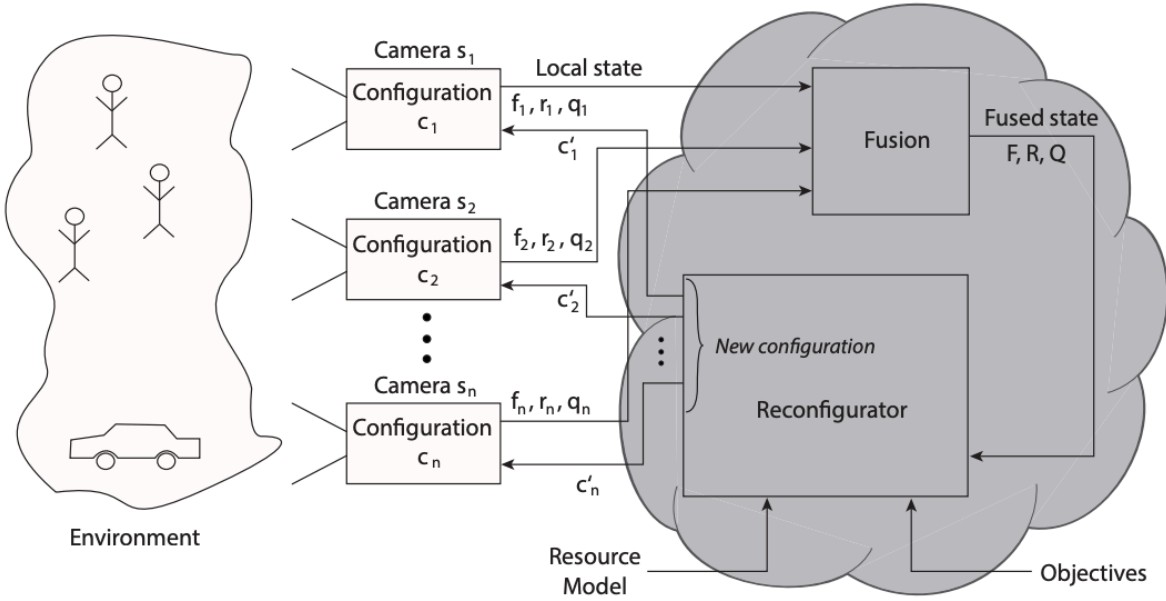

**Figure 3.** Self-reconfiguration framework as proposed in [8].

This article proposes an adaptive self-reconfiguration (ASR) framework to extend the scope of the framework proposed by Piciarelli et al. [8] to a number of networks, such that each network is configured to learn and reconfigure adaptively by utilizing the experiences and models of others to obtain the best visual understanding possible and thus perform optimal reconfiguration of the SCN. In spite of using a centralized reconfigurator, we propose a distributed network of systems comprising a number of datacenters or cloud servers to perform data computation and reconfiguration. For illustration, the architecture is inspired by the NEAR blockchain network, comprising a number of datacenters, such that each datacenter is capable of sharing datasets, performance parameters, and even trained models. Due to the symmetrical architecture of the distributed blockchain network, the proposed ASR system provides unbiased functionality for each smart camera network utilizing the network. This allows the model, datasets, and parameters to be used by any SCN to deal with an unforeseen situation if a similar condition has been encountered and dealt with by another SCN in the distributed network. Further, the ASR framework proposes the distribution of critical data throughout the blockchain network, providing data security, such that if an SCN is attacked by adversarial attack, the critical data can be retrieved. In some respects, the network of systems developed or deployed using the ASR framework utilizing a distributed network (blockchain) can be considered a self-adaptive system of active vision systems.

*Model*

An exemplary embodiment of the proposed ASR-framework-based architecture for deployment of the adaptive self-reconfiguration of active vision systems utilizing SCNs is shown in Figure 4. The framework consists of "m" number of active vision systems (AVSs) connected together in a distributed network. The architecture comprises "m" number smart camera networks coupled to each other by way of a distributed blockchain network "B". Sensed data and a local configuration space are associated with each sensor node of each SCN, which results in input data from each sensor being processed for reconfiguration by one or more datacenters in the blockchain network. Input data, local states, resources, and quality of service information from each sensor "$s_i$" are cumulatively passed to a fusion block of the SCN (represented by the set {Ci}) to obtain an overall system configuration space for the SCN. Each SCN has an overall functionality, defined by a number of objectives. The sensed data, the system's configuration space, and the SCN objectives are communicated as self-expression data of the SCN "$E_i$" to the blockchain network "B" for processing.

The blockchain network comprises "M" number of datacenters (cloud servers) that are configured to perform computations on the self-expression data ($E_i$) for each SCN. A datacenter is selected as per the objectives of the SCN, based on the resources available at the datacenter. The datacenter generates a model based on the objectives of the SCN and derives an understanding by way of the sensor's data using unsupervised artificial intelligence. Each activity and/or event determined by a model (by the datacenter) is assigned a pattern vector for identification, such that the pattern vector is generated based on the objectives and the understanding of the event by the model. The model further derives a best-possible configuration space corresponding to the event iteratively. Each pattern vector along with the model specifications is distributed to every datacenter in the blockchain network "B". Each event detected by any datacenter generates a pattern vector that is distributed throughout the entire blockchain "B". Based on the understanding of the detected activity, the selected datacenter generates self-awareness data ($A_i$) comprising a revised configuration space set $\{C_i'\}$ for the SCN. Based on the revised configuration space set, each camera sensor ($s_{ij}$) is calibrated.

If an unforeseen condition is faced by another SCN in the network, the associated datacenter generates a pattern vector and compares the pattern vector with a pre-existing pattern vector in the blockchain network. Based on the comparison of the pattern vector with the pre-existing pattern vectors, a suitable datacenter is allotted and thus the latency of iterative processing is minimized. The improved configuration space is reverted back to the SCN through self-awareness data that are utilized by the reconfigurator to calibrate the parameters of each sensor of the SCN.

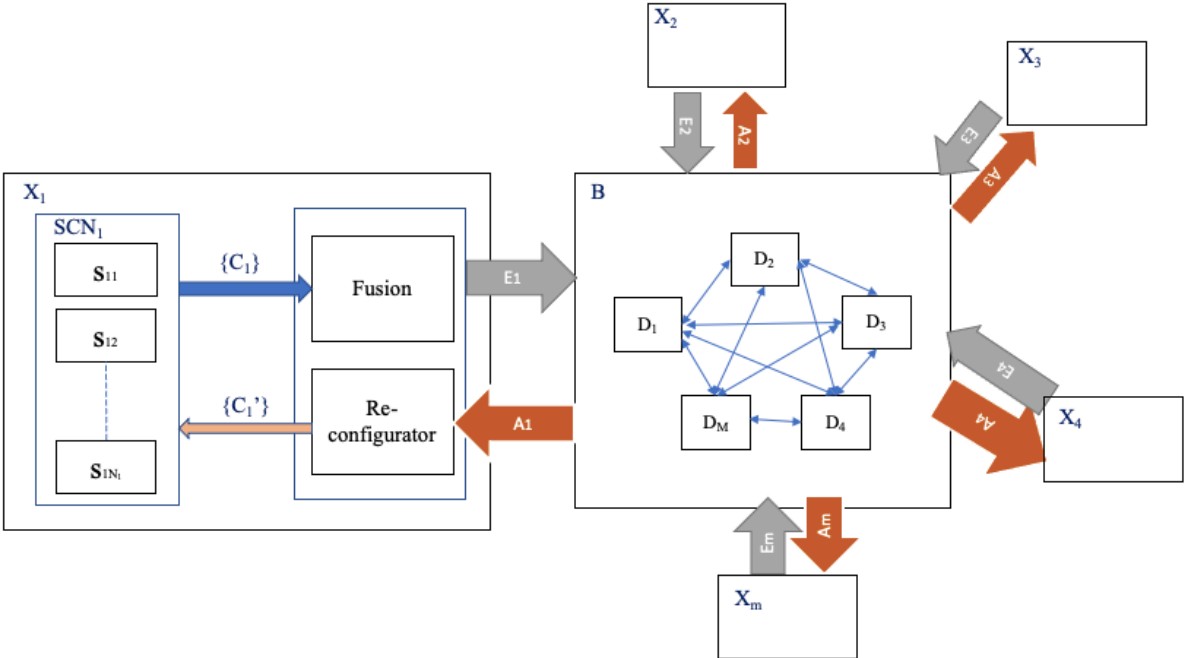

**Figure 4.** Adaptive self-reconfiguration framework for SCN-enabled active vision systems.

Notation used:
$X_i$: ith smart camera network;
$s_{ij}$: jth camera sensor of the ith active vision system;
$\{C_i\}$: Set of input data for the ith SCN;
$\{C_i'\}$: Set of output data for the ith SCN;
$E_i$: Self-expression data for the ith SCN;
$A_i$: Self-awareness data for the ith SCN;
$n_i$: Number of camera sensors in the ith active vision system;
$m$: Number of active vision systems utilizing the ASR framework;

B: Distributed blockchain network comprising "M" number of datacenters;

$D_k$: kth datacenter in the distributed blockchain network.

To generate the pattern vector corresponding to each event, we propose an auto-encoder-based unsupervised learning model, such that, for each event, the sensor data received by a selected datacenter is converted to lower-dimensional data. The auto-encoder model determines a reconstruction error to restore the sensor data back to their original form. The reconstruction error is propagated to the auto-encoder to iteratively alter the weights of the model for training in order to minimize the reconstruction error. The data of weights of the trained model along with the objectives of the SCN are utilized to generate the pattern vector, such that the pattern vector is distributed throughout the blockchain. The datacenter further computes the reconfiguration parameters based on the understanding of the event and sends the reconfiguration parameters to the SCN in the form of self-expression data, which are utilized by the reconfigurator of the SCN to calibrate the parameters. If an identical event occurs in any other SCN in the blockchain, the selected datacenter generates a pattern vector for the event. The pattern vector is matched with the pre-existing pattern vectors in the blockchain, and the reconfiguration model of the closest pattern vector is shared with the SCN.

For an unbiased and seamless flow of operations, the distributed blockchain network protocols were designed based on consensus mechanisms. The datacenters are categorized as processor datacenter nodes and validator datacenter nodes. The categorization of the datacenters is based on the proof-of-active-participation (POAP) consensus mechanism [180]. The processor datacenter nodes generate a model and a pattern vector based on the sensor data, whereas the validator nodes validate the model and the pattern vectors. The processor nodes further share details of their resource consumption (bandwidth and computational resources, etc.) to obtain rewards. The validator nodes validate or authenticate the details about resource consumption, models, and pattern vectors generated by the processor nodes. The validator nodes further compare the pattern vector generated by the processor node with the pre-existing pattern vectors to determine the best suitable reconfiguration configurations. To ensure the unbiased functionality of the validator and the processor nodes, each participating processor node and validator node is required to raise a stake based on a proof-of-stake consensus mechanism [181]. Upon validation of the model and the reconfiguration, the processor nodes and the validator nodes receive their stake as well as a reward from the corresponding SCN network. A flow diagram of the process is shown in Figure 5.

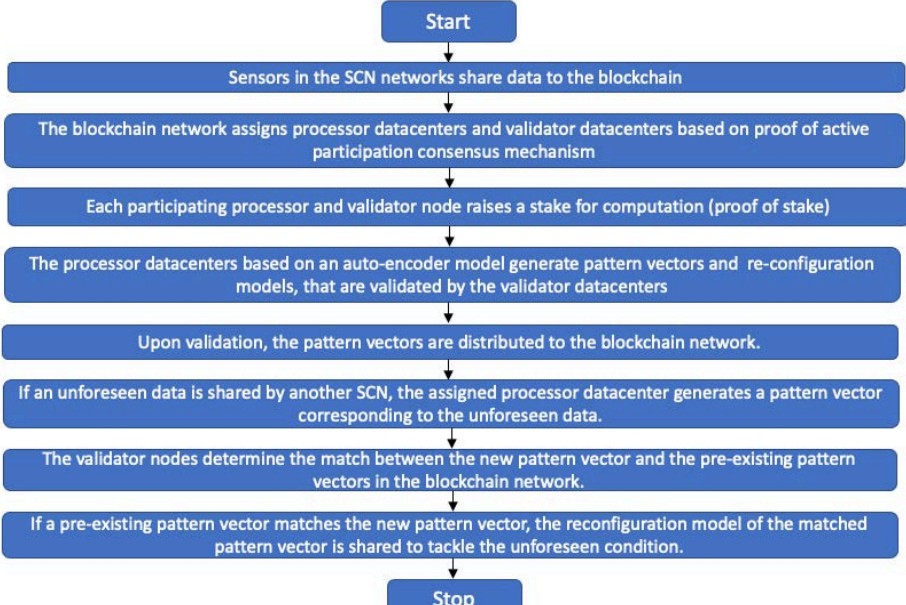

**Figure 5.** Flow diagram illustrating the adaptive reconfiguration using the ASR framework.

## 7. Results

For illustration of the functionality of the adaptive self-reconfiguration (ASR) framework, we utilized multiple surveillance video datasets and simulated results, comparing the centralized reconfiguration of [8] to our proposed distributed adaptive self-reconfiguration. Each frame of each video dataset was standardized to a resolution of 640 × 360 pixels. Activity maps were generated using the centralized approach presented in [8] and the proposed ASR, respectively, such that the activity maps updated with each frame added to an event. The performances of the systems were compared in terms of multi-object tracking accuracy (MOTA) results obtained for both activity maps. Further, due to the unavailability of resources necessary to develop a private blockchain network with datacenters, we utilized lower processing capabilities and thus, in spite of accurate processing latency, the results are represented with respect to training cycles (Ti) (each of 15 minutes duration); however, the processing capabilities of a datacenter are much higher and can be utilized to achieve results in near real time. We utilized standard regions with a convolutional neural network (R-CNN) model for multi-object detection by the centralized system of [8] as well as the distributed system based on the proposed ASR framework. All the results presented hereinbelow were derived using the MatLab Image Processing Toolbox. The activity map corresponding to the proposed ASR was further utilized to predict the upcoming event in the next frame. The SAR framework was utilized for multi-object detection by the exemplary embodiment method described in Section 6; however, it can be utilized to enhance any performance parameter, as illustrated in Section 2.

True-positive pixel count (TPC), false-positive pixel count (FPC), false-positive pixel-detection rate (FPR), true-negative pixel count (TNC), true-negative pixel-detection rate (TNR), and false-negative pixel count (FNC) were used as primary performance parameters to obtain MOTA values using Equation (1).

$$\text{MOTA (\%)} = (\text{Total count of pixels} - \text{falsely detected pixels}) \times 100 \tag{1}$$

### 7.1. Surveillance Dataset 1

A comparison of the performance of the system presented in [8] and the proposed ASR system, tested on surveillance dataset 1 in terms of MOTA, is presented in Table 9 and Figure 6. The predictions of the directions of vehicle movement in various random frames from surveillance dataset 1 based on the generated activity map are presented in Figure 7.

**Table 9.** Comparison of the performance for the system in [8] and the proposed ASR for surveillance dataset 1.

| Pixels: 640 × 360 | | T1 | T2 | T3 | T4 | T5 | T6 | T7 |
|---|---|---|---|---|---|---|---|---|
| [8] | TPC | 30,541 | 41,556 | 52,956 | 56,871 | 60,279 | 74,638 | 76,267 |
| | TNC | 18,719 | 20,268 | 28,514 | 33,400 | 39,277 | 35,098 | 41,905 |
| | FPC | 80,271 | 74,352 | 69,183 | 66,418 | 63,116 | 60,947 | 59,104 |
| | FNC | 1,00,869 | 94,024 | 79,747 | 73,711 | 67,728 | 59,717 | 53,124 |
| | **MOTA (%)** | **21.38** | **26.92** | **35.36** | **39.18** | **43.21** | **47.63** | **51.29** |
| | TRAINING CYCLES TO OBTAIN ABOVE 80% MOTA: **18** | | | | | | | |
| ASR | TPC | 63,018 | 69,217 | 72,141 | 76,238 | 78,908 | 79,519 | 86,211 |
| | TNC | 33,128 | 37,320 | 41,169 | 46,473 | 48,042 | 52,793 | 51,775 |
| | FPC | 47,982 | 43,755 | 40,073 | 36,117 | 36,431 | 31,824 | 29,273 |
| | FNC | 86,272 | 80,108 | 77,017 | 71,512 | 67,019 | 66,264 | 63,141 |
| | **MOTA (%)** | **41.73** | **46.24** | **49.18** | **53.26** | **55.10** | **57.34** | **59.89** |
| | TRAINING CYCLES TO OBTAIN ABOVE 80% MOTA: **12** | | | | | | | |

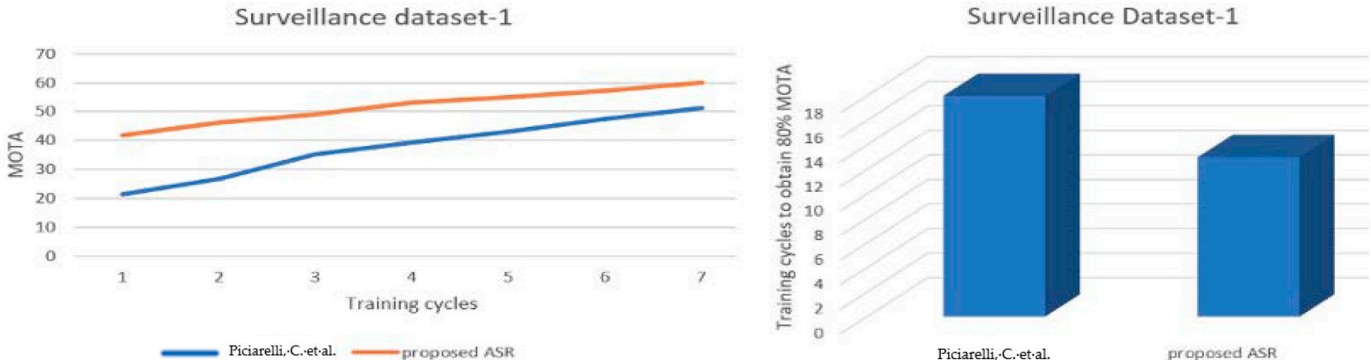

**Figure 6.** Comparison of the performances of the system in [8] and the proposed ASR for surveillance dataset 1.

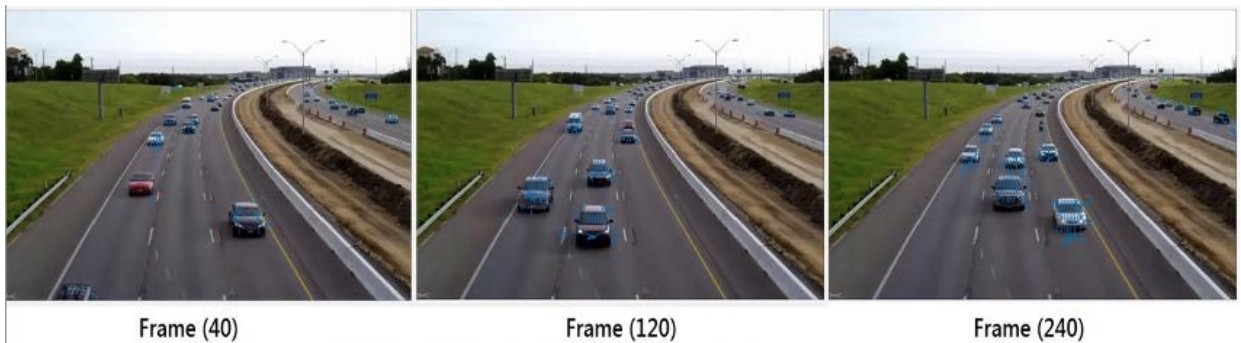

**Figure 7.** Predicted motion of multiple objects in random frames from surveillance dataset 1.

*7.2. Surveillance Dataset 2*

A comparison of the performances of the system in [8] and the proposed ASR, tested on surveillance dataset 2 in terms of MOTA, is presented in Table 10 and Figure 8. The predictions of the directions of vehicle movement in various random frames from surveillance dataset 2 based on the generated activity map are presented in Figure 9.

**Table 10.** Comparison of the performances of the system in [8] and the proposed ASR for surveillance dataset 2.

|  |  | T1 | T2 | T3 | T4 | T5 | T6 | T7 |
|---|---|---|---|---|---|---|---|---|
| [8] | TPC | 23,211 | 27,324 | 29,841 | 33,266 | 36,421 | 39,972 | 41,101 |
|  | TNC | 50,080 | 58,477 | 66,581 | 69,515 | 76,451 | 77,601 | 83,591 |
|  | FPC | 89,233 | 85,161 | 82,686 | 79,957 | 75,277 | 72,098 | 69,035 |
|  | FNC | 67,876 | 59,438 | 51,292 | 47,662 | 42,251 | 40,729 | 36,673 |
|  | **MOTA (%)** | **31.81** | **37.24** | **41.85** | **44.61** | **48.99** | **51.03** | **54.12** |
|  | TRAINING CYCLES TO OBTAIN ABOVE 80% MOTA: **15** | | | | | | | |
| ASR | TPC | 38,211 | 40,128 | 41,007 | 42,091 | 43,108 | 43,236 | 43,901 |
|  | TNC | 70,860 | 77,652 | 83,017 | 84,952 | 90,317 | 92,884 | 96,666 |
|  | FPC | 77,102 | 71,928 | 69,982 | 67,041 | 66,101 | 65,384 | 63,687 |
|  | FNC | 44,227 | 40,692 | 36,394 | 36,316 | 30,874 | 28,896 | 26,146 |
|  | **MOTA (%)** | **47.34** | **51.12** | **53.83** | **55.14** | **57.91** | **59.08** | **61.01** |
|  | TRAINING CYCLES TO OBTAIN ABOVE 80% MOTA: **11** | | | | | | | |

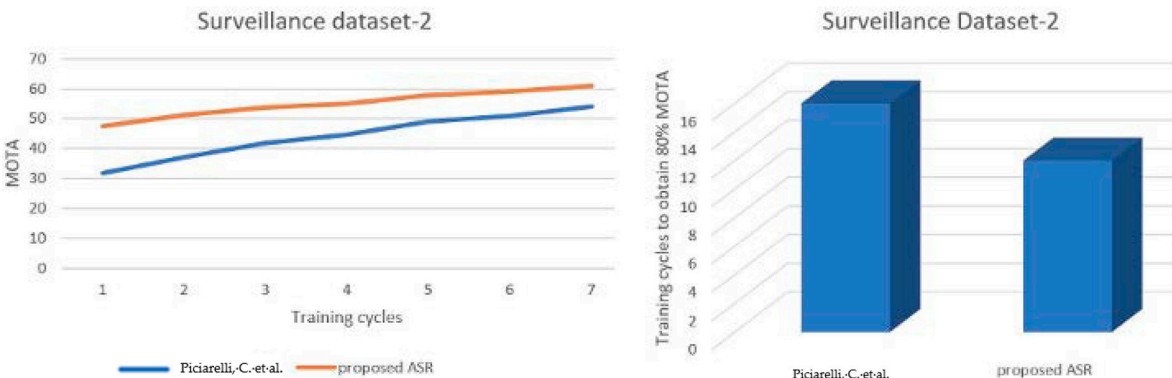

**Figure 8.** Comparison of the performances of the system in [8] and the proposed ASR for surveillance dataset 2.

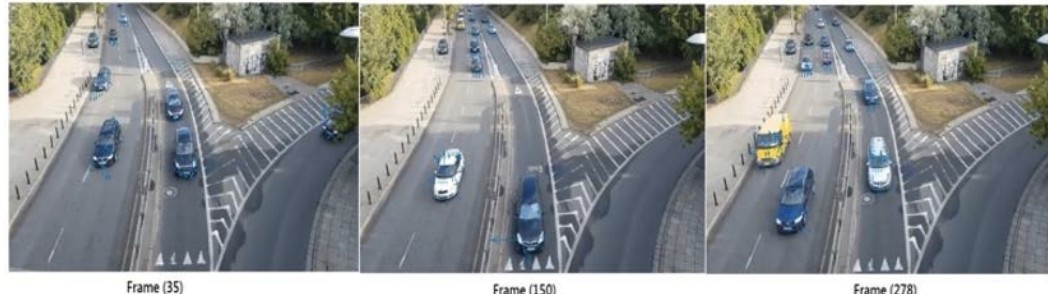

**Figure 9.** Predicted motions of multiple objects in random frames from surveillance dataset 2.

A comparison of the performances of the system presented in [8] and the proposed ASR for multi-object tracking applications are represented in Figures 6 and 8. It must be noted that, as the reconfiguration is dependent on tracking accuracy, the reconfiguration presented by the distributed ASR model is better than that of the centralized reconfigurator model of [8].

## 8. Conclusions and Scope

The performance of active vision systems and reconfiguration of the sensors providing data to active vision systems are interdependent. However, the reconfiguration of the calibration space of an active vision system employing a network of sensors can be challenging. Most of the state-of-the-art active vision systems fail miserably in dealing with unforeseen conditions, as the reconfiguration model takes time to adapt to the new conditions to develop an understanding of an unforeseen event. Thus, reconfiguring such a system in real time is nearly impossible. Further, most active vision systems nowadays rely on artificial-intelligence-based models for the processing of sensor data to develop an understanding of an event. However, such models are prone to adversarial attacks and are thus threatened with data loss. Therefore, such systems cannot be relied on for making critical decisions in real time.

This article discusses the challenges at different operational levels in deploying an active vision system employing a camera network. This article presents a detailed description of the systems and methods proposed for addressing the challenges with respect to both data processing and reconfiguration, along with the state-of-the-art solutions for the same, proposing an adaptive self-reconfiguration (ASR) framework employing a blockchain-based distributed network for data processing and reconfiguration of the sensor network's configuration space. To make the understanding of the framework easier, this article has briefly defined the concepts of self-adaptation and self-reconfiguration in systems prior to the ASR framework.

The blockchain network of the ASR-framework-based architecture acts as a system of systems, connecting a number of smart camera networks together in a distributed

architecture. The blockchain further includes a number of datacenters configured to obtain data from the SCNs and perform data processing to obtain understanding about the events and activities in the scene. The datacenters further generate pattern vectors corresponding to the event or activity detected and distribute pattern vectors along with corresponding reconfiguration models to each datacenter in the distributed blockchain network, such that if a similar event is observed at any other SCN in the network, the reconfiguration model associated with a pre-existing pattern can be utilized for a much faster reconfiguration of the SCN. The blockchain network is founded on proof-of-stake and proof-of-active-participation consensus mechanisms to maintain an unbiased and smooth flow of operations between all the participating datacenters in the network. Further, due to the distributed architecture, critical data float in the blockchain network and thus the effect of adversarial attacks is minimized.

This article further compared the performance of a centralized active vision system with a distributed system based on the proposed ASR framework for multi-object tracking and showcased enhanced tracking performance in terms of multi-object tracking accuracy and low latency. The proposed framework was tested on a homogeneous environment, with some limitations and assumptions; however, the ASR framework aims to be developed for heterogeneous systems to enhance the scope of its applications in future.

**Author Contributions:** S.: Lead author of the manuscript (corresponding author), conceptualization and methodology, writing—original draft preparation, investigation, and editing; I.S.: Second author, research design, guidance, and reviewing. All authors have read and agreed to the published version of the manuscript.

**Funding:** This research received no external funding.

**Data Availability Statement:** Training of both models (i.e., for [8] and the proposed ASR) was performed on a standard surveillance video dataset. The models were further tested on surveillance dataset 1 and surveillance dataset 2. The standard surveillance dataset, as well as surveillance dataset 1 and surveillance dataset 2, are non-restricted and free-to-use third-party datasets with $640 \times 360$-pixel resolutions. The datasets and the training and testing models can be accessed by clicking on this link: https://drive.google.com/drive/folders/1ZizuTfDZ2gT-ro1o2lAAo_SVsw4_MY-7 (accessed on 23 October 2022).

**Acknowledgments:** This work was carried out under the supervision of Indu Sreedevi in the Department of ECE, Delhi Technological University, New Delhi, India, and Shashank expresses immense gratitude to his guide and UGC for enlightening him throughout the process.

**Conflicts of Interest:** The authors declare that they have no conflicts of interest. The authors declare that they have no known competing financial interests or personal relationships that could have appeared to influence the work reported in this article.

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
