# Peer review of "Distributed Network of Adaptive and Self-Reconfigurable Active Vision Systems"

_symmetry, doi:10.3390/sym14112281_

Round 1
Reviewer 1 Report
The paper considers active computer vision systems. Ways to improve their various parameters are proposed. Of primary interest are blockchain ideas for SCN configuration. There are quite a few comments on the article:
1) The level of language in the article immediately catches the eye. Instead of "paper" should be used "article". I would recommend going through proof-reading.
2) In the review part, among the shortcomings, it is worth noting such a problem as visual attacks ( https://www.mdpi.com/2076-3417/11/11/5235/pdf?version=1622826749) It is found among the authors, but not sufficiently disclosed. It is recommended to refer to the sources provided.
3) In the text of the article, links should go in order. At the same time, when considering the tasks and trends of computer vision, it is necessary to refer to review works (DOI: 10.18287/2412-6179-CO-922)
4) It would be nice to accompany such large pieces of review text with graphics. For example, visualize classifications of computer vision problems, the growing popularity of this field, types of detectors, etc.
5) You can not separate the pages of figures, tables and captions to them. For example, this is in figure 4.
6) Formula (1) is not designed in style.
7) It would be interesting to compare with the results of YOLO + Optical Flow.
Author Response
Respected Reviewer,
Greetings.
Thanks for sharing your valuable comments and suggestions on our manuscript and contributing in the revision of our paper.
In reference to your comments, kindly take our responses in consideration:
- The level of language in the article immediately catches the eye. Instead of "paper" should be used "article". I would recommend going through proof-reading.
We have tried to revise the manuscript to enhance the ease of readability. We have further tried to improve the paper grammatically as well. As per your suggestion, we have used “article” in place of using paper.
- In the review part, among the shortcomings, it is worth noting such a problem as visual attacks ( https://www.mdpi.com/2076-3417/11/11/5235/pdf?version=1622826749) It is found among the authors, but not sufficiently disclosed. It is recommended to refer to the sources provided.
We have revised the review part including visual attacks. We have added some references for the same including the suggested reference.
- In the text of the article, links should go in order. At the same time, when considering the tasks and trends of computer vision, it is necessary to refer to review works (DOI: 10.18287/2412-6179-CO-922)
The paper has been revised such that the references go in order. We have referred to some references wherever required including the suggested reference.
- It would be nice to accompany such large pieces of review text with graphics. For example, visualize classifications of computer vision problems, the growing popularity of this field, types of detectors, etc
To make the long texts easier to read and draw a trend of the review part, we have incorporated year wise tables of proposed solutions along with their advantages.
- You can not separate the pages of figures, tables and captions to them. For example, this is in figure 4.
We have tried to take care of the abovementioned comment in the revised paper.
- Formula (1) is not designed in style.
Equation 1 has been revised as per the symmetry template.
- It would be interesting to compare with the results of YOLO + Optical Flow.
Yolo+ optical flow are great for advanced multi-object detection systems. However, the proposed ASR framework is not limited to enhancement of object detection or tracking efficiency. The overall objective of the proposed ASR is to showcase a framework for development of adaptive and self-reconfigurable active vision system. Multi-object detection has been taken As an exemplary embodiment of the ASR.
We have trained and tested the ASR framework on standard R-CNN using MatLab image processing tool suite. However, we will work on training and testing the ASR framework on YOLO + optical flow too.
We hope, that we could address all the comments and suggestions provided by you for publication of our paper.
Thanks a lot for your efforts and suggestions.
Sincerely,
Shashank.

Reviewer 2 Report
The presented manuscript has an above average number of pages and provides an impressive number of references. I see two potentially significant contributions of the manuscript; however, both need improvement.
The first contribution is a literature review of the challenges and related solutions for active vision systems, given in sections 3 to 5. The review covers an impressive number of publications however to be considered a serious systematic literature review, a section describing used methodology has to be added. Such section should provide details, such as used publication or citation databases, years covered and search queries used. If this was not an intention of the authors and sections 3 to 5 only provide related work for the next part of the manuscript, then these sections should be reduced and focus only on the references directly related to that next part.
The second contribution (sections 6-7) is an improvement of the reference framework from the paper cited as [148] and experimental evaluation of that improvement. The improvement dwells in extending the reference model with an adaptive self-reconfiguration framework. The first question here is why the authors decided to improve a reference model that is 7 years old. The second issue is that the authors do not make it easy for a reader to compare their improved model (Figure 3) with the original model (Fig.1 in [148]) as they introduce different symbols to designate parts of the model. I suggest the authors to use the same symbols (letters, indices) to designate the parts of the model and also present their model in a visually similar way to the one in Fig.1 in [148]. The third issue is a lack of details on the implementation of both models in section 7, as both models are quite abstract. It will be best to provide a link from which the models and datasets can be downloaded to be able to replicate the experiment. It should also be explained, in the text of Section 7, what T1 to T7 mean.
There are also other, minor, issues in the manuscript that should be addressed. Rather important one is an almost absolute lack of references in sections 1 and 2, which introduce important terms and context for the manuscript. The authors should also improve the English by correcting typos and modifying awkward statements, which are quite common in the manuscript. For example:
lines(l.) 16-17: The contribution through this paper l.19: discusses about the role l.81: computer vision system employing a smart camera network (SCN) l.104: higher information l.139-140: frame-139 to-pixel relationship l.772: dimention l.774: propogated l.817: We further submit l.863: discusses about
Author Response
Respected Reviewer,
Greetings.
Thanks for sharing your valuable comments and suggestions on our manuscript and contributing in the revision of our paper.
In reference to your comments, kindly take our responses in consideration:
- The first contribution is a literature review of the challenges and related solutions for active vision systems, given in sections 3 to 5. The review covers an impressive number of publications however to be considered a serious systematic literature review, a section describing used methodology has to be added. Such section should provide details, such as used publication or citation databases, years covered and search queries used. If this was not an intention of the authors and sections 3 to 5 only provide related work for the next part of the manuscript, then these sections should be reduced and focus only on the references directly related to that next part.
As per your suggestion, we have added a sub-section under section 3 providing details of the citation databases, years covered and search queries used.
- The second contribution (sections 6-7) is an improvement of the reference framework from the paper cited as [148] and experimental evaluation of that improvement. The improvement dwells in extending the reference model with an adaptive self-reconfiguration framework. The first question here is why the authors decided to improve a reference model that is 7 years old. The second issue is that the authors do not make it easy for a reader to compare their improved model (Figure 3) with the original model (Fig.1 in [148]) as they introduce different symbols to designate parts of the model. I suggest the authors to use the same symbols (letters, indices) to designate the parts of the model and also present their model in a visually similar way to the one in Fig.1 in [148]. The third issue is a lack of details on the implementation of both models in section 7, as both models are quite abstract. It will be best to provide a link from which the models and datasets can be downloaded to be able to replicate the experiment. It should also be explained, in the text of Section 7, what T1 to T7 mean.
The earlier reference [148] proposed self-reconfiguration framework for smart camera network and is the reference framework for some of the recent and most advanced active vision systems such as:
- Cai, Liangliang, Hanyuan Ma, Zhuocheng Liu, Zhaoxin Li, and Zhong Zhou. "Coverage Control for PTZ Camera Networks Using Scene Potential Map." In 2022 IEEE International Conference on Multimedia and Expo (ICME), pp. 1-6. IEEE, 2022; and
- Suresh, Sumi, and Vivek Menon. "An Efficient Graph Based Approach for Reducing Coverage Loss From Failed Cameras of a Surveillance Network." IEEE Sensors Journal 22, no. 8 (2022): 8155-8163.
We further wish to highlight that none of the present day self-reconfiguration systems present an adaptive learning approach in a distributed network.
We have tried to compare and relate our improved model with the original model (Fig.1 in [148]) however, it must be noted that the proposed ASR framework is inspired by the functionality of the self-reconfiguration framework and therefore does not entirely rely on the operation and functionality of the centralized self-reconfiguration framework as proposed in [148] (earlier reference).
The overall objective of the proposed ASR is to showcase a framework for development of adaptive and self-reconfigurable active vision system. Multi-object detection has been taken As an exemplary embodiment of the ASR. The proposed ASR framework is not limited to enhancement of object detection or tracking efficiency only. We have trained and tested the ASR framework on standard R-CNN using MatLab image processing tool suite.
The training and testing datasets, as well as the R-CNN models can be accessed through the following link: link.
T1-T7 represent training cycles. As we have limited computational resources, therefore, the performance of a datacenter can-not be matched replicated. Therefore, we have presented the multi-object tracking accuracy in terms of training cycles (Ti). It has also been added in the revised paper.
- There are also other, minor, issues in the manuscript that should be addressed. Rather important one is an almost absolute lack of references in sections 1 and 2, which introduce important terms and context for the manuscript. The authors should also improve the English by correcting typos and modifying awkward statements, which are quite common in the manuscript. For example: lines(l.) 16-17: The contribution through this paper l.19: discusses about the role l.81: computer vision system employing a smart camera network (SCN) l.104: higher information l.139-140: frame-139 to-pixel relationship l.772: dimention l.774: propogated l.817: We further submit l.863: discusses about
We have added references as per your kind suggestions and wherever required. We have also made all the necessary corrections as per your suggestions.
We hope, that we could address all the comments and suggestions provided by you for publication of our paper.
Thanks a lot for your efforts and suggestions.
Sincerely,
Shashank.

Round 2
Reviewer 1 Report
The authors eliminated all the shortcomings. High quality of work is provided.
Reviewer 2 Report
Although the revised version did not address all the remarks given in the review of the first version, the changes have been significant enough to consider the manuscript acceptable.
However, my personal opinion is that the authors may capitalize on their effort much better if they split the manuscript into two ones, the first dealing with the literature review and the second one with the improved SCN.